Yuanansuchus maopingchangensis sp. nov., the second capitosauroid temnospondyl from the Middle Triassic Badong Formation of Yuanan, Hubei, China

Liu Jun liujun@ivpp.ac.cn
Key Laboratory of Vertebrate Evolution and Human Origins of Chinese Academy of Sciences, Institute of Vertebrate Paleontology and Paleoanthropology, Chinese Academy of Sciences , China
Sues Hans-Dieter
Electronic publication date: 2016 Apr 5
Publication date: 2016
Volume: 4
Electronic Location ID: e1903
Received 2015 Nov 19; Accepted 2016 Mar 15
Copyright: ©2016 Liu
Copyright year: 2016
Copyright holder: Liu
License: This is an open access article distributed under the terms of the Creative Commons Attribution License, which permits unrestricted use, distribution, reproduction and adaptation in any medium and for any purpose provided that it is properly attributed. For attribution, the original author(s), title, publication source (PeerJ) and either DOI or URL of the article must be cited.
License URL: https://creativecommons.org/licenses/by/4.0/

Keywords: Triassic, Anisian, Badong Formation, Temnospondyli, Capitosauria, Capitosauroidea

Funding: National Basic Research 2012CB821902 National Natural Science Foundation of China 41472017 This study is supported by National Basic Research (973) project (2012CB821902) and National Natural Science Foundation of China grants (41472017). The funders had no role in study design, data collection and analysis, decision to publish, or preparation of the manuscript.

==============================
A second species of Yuanansuchus, Y. maopingchangensis, is erected for new temnospondyl material from the Maopingchang site, Yuanan County, Hubei Province, China. These specimens are from the same horizon that produced Y. laticeps, the type species. Y. maopingchangensis shares the following features with Y. laticeps: postorbital portion of skull occupied more than 1/3 of skull length; tabular horn directed laterally; lateral line sulci continuous, well impressed; supraorbital sensory canal enters lacrimal; preorbital projection of jugal less than half length of snout; and vomerine plate short. However, Y. maopingchangensis differs from Y. laticeps in having an elongated skull, frontal extending posteriorly to the level of the posterior orbital margin, closed otic fenestra, cultriform process of parasphenoid extending to the level of the anterior margin of the interpterygoid vacuities, and absence of fodina vomeralis. Analysis of a new capitosaur phylogenetic data matrix, containing 56 characters and 29 species, confirms that the two species from Yuanan are sister taxa and that Capitosauroidea can be divided into two major clades: one including Parotosuchus, Eryosuchus, Calmasuchus and Cherninia, and another including Mastodonsaurus, Yuanansuchus, Stanocephalosaurus, Procyclotosaurus, Paracyclotosaurus, Antarctosuchus, Eocyclotosaurus, Quasicyclotosaurus, Tatrasuchus and Cyclotosaurus.

Introduction

Temnospondyls are the largest and most speciose group of amphibians. Within this group, the stereospondyl clade Capitosauria produced some of the largest species and dominated many Mesozoic aquatic ecosystems (e.g., Schoch, 2008). Capitosaur research has a long history, and more than 120 nominal species have been erected within this group (Damiani, 2001; Fortuny, Galobart & De Santisteban, 2011; Liu & Wang, 2005; Sidor, Steyer & Hammer, 2014; Sulej & Niedźwiedzki, 2013). The taxonomy of Capitosauria was poorly understood for a long time, but recent systematic revisions of the group by Schoch & Milner (2000) and Damiani (2001) have led to a consensus regarding the taxonomic status of most previously named genera and species. Meanwhile, large-scale cladistic analyses of Capitosauria have been undertaken by Schoch (2000) and Damiani (2001). To resolve the position of Yuanansuchus laticeps, Liu & Wang (2005) added Y. laticeps and Quasicyclotosaurus campi to the matrix of Damiani (2001). In later works, researchers modified the character list and data matrix of Damiani (2001) for their own studies (Fortuny, Galobart & De Santisteban, 2011; Schoch, 2008; Sidor, Steyer & Hammer, 2014), and they produced the phylogenetic hypotheses that have varied widely in such particulars as the position of Edingerella, the proximity of Eryosuchus to Mastodonsaurus, and the content of the subgroup Capitosauroidea (sensu Schoch, 2013) (Fig. 8 of Schoch (2008) vs. Fig. 7 of Fortuny, Galobart & De Santisteban (2011)).

Capitosaurs are widely distributed on the land masses derived from the Pangean supercontinent, and are known from many well-preserved specimens (e.g., Damiani, 2001; Schoch & Milner, 2000). However, this group is poorly represented in east Asia (Ingavat & Janvier, 1981; Li, Wu & Zhang, 2008), a region from which only two highly informative capitosaur specimens have been described: the posterior part of a skull almost identical to the corresponding cranial region of Cyclotosaurus posthumus was reported from Thailand (Ingavat & Janvier, 1981), and the nearly complete skull of a taxonomically novel capitosaur was reported from the middle Triassic Badong Formation of China (Liu & Wang, 2005). The notably broad Chinese skull was collected in Maopingchang Township, Yuanan County, Hubei Province in 2001, and was named Yuanansuchus laticeps (Liu & Wang, 2005). In 2011, a mandibular fragment was reported from the Lower Triassic of Japan, implying that the dispersal of stereospondyls into China occurred before the latest Early Triassic (Nakajima & Schoch, 2011).

In 2011, my team made a second trip to the Maopingchang locality and collected a considerable number of fossils. In this paper, several new temnospondyl specimens are described and assigned to a new species, Yuanansuchus maopingchangensis. Y. maopingchangensis represents the second temnospondyl species from Maopingchang. To evaluate the phylogenetic relationships of these Chinese capitosauroids, a new character list is established based on a revision of previously established characters, and new matrices including both Y. laticeps and Y. maopingchangensis are constructed.

Geological Setting

Outcrops of the Triassic Badong Formation are widely distributed in Hubei, Chongqing, and Hunan. The Badong Formation conformably overlies Lower Triassic shallow marine carbonates of the Jialinjiang Formation, and in most places underlies an Upper Triassic terrestrial coal series. The complete Badong Formation is generally divided into five members (numbered I to V), but whole five members only present in a few areas such as Badong, Sangzhi. In Yuanan, only memebers I to IV are present as showing by the geological map and lithological column (Fig. 1). Vertebrate fossils have been collected from the purplish-red calcareous siltstone and mudstone of Member II in both Hunan and Hubei (Liu & Wang, 2005; Zhang, 1975). In Yuanan, the bonebed lies in the middle portion of the Member II (Fig. 1).

Figure 1 (A) A simplified geological map of Maopingchang area to show the fossil locality (red star in the map); (B) lithological column of Badong Formation showing the relative position of fossil horizon (from Yuanan Geological map, 1/200000). Abbreviations: B1-4, member 1-4 of Badong Formation; J, Jianlingjiang Formation; X, Xiangxi Group.

Member I contains a rich bivalve fauna, which is generally regarded as Anisian in age. Member III yields the bivalves Plagiostoma striatum and Placunopsis plana, the ammonoid Progonoceratites, and the conodont Neospathodus kockeli, and is accepted as Anisian in age although Zhang, Chen & Palmer (2003) argued that it could instead be Ladinian. Member II has also been suggested to be Anisian in age, based on its fossil flora (Meng, Zhang & Xu, 1995), so the fossil tetrapods from Member II are accepted here as Anisian, possible late Anisian, as also suggested by magnetostratigraphic work (Huang & Opdyke, 2000). Member II has been interpreted as comprising tidal flat sediments (Meng, Zhang & Xu, 1995), but the sediments at the Maopingchang locality are more likely to represent a floodplain facies. No tidal-related structures were observed in the field.

The specimens described in this paper were excavated from a hill 300 m away from the type locality of Yuanansuchus laticeps (Fig. 2). The new locality is approximately the same stratigraphic level as that of Y. laticeps on the section. This fossiliferous layer is less than 10 m in thickness. Among the bones collected at this new spot, four skulls, one left lower jaw, three interclavicles, and nine clavicles are of temnospondyl origin. They represent at least six individuals, based on the number of clavicles. In addition to the temnospondyl fossils from the type locality of Y. maopingchangensis, vertebrae with high neural spines were also discovered and almost certainly belong to an archosaur similar to Lotosaurus from Sangzhi County in Hunan (Zhang, 1975).

Figure 2 Fossil locality.

The fossil locality at Maopingchang Township, Yuanan County, Hubei Province, China. All described specimens came from this locality, and the blue arrow points to the place of the holotype. Photo credit: Jun Liu.

Nomenclatural acts—The electronic version of this article in Portable Document Format (PDF) will represent a published work according to the International Commission on Zoological Nomenclature (ICZN), and hence the new names contained in the electronic version are effectively published under that Code from the electronic edition alone. This published work and the nomenclatural acts it contains have been registered in ZooBank, the online registration system for the ICZN. A ZooBank LSID (Life Science Identifier) can be resolved and the associated information viewed through any standard web browser by appending the LSID to the prefix http://zoobank.org/. The LSID for this publication is: urn:lsid:zoobank.org:pub: D68F2D09-0AC0-4AE1-ACE1-EFD0C5680916. The online version of this work is archived and available from the following digital repositories: PeerJ, PubMed Central and CLOCKSS.

Systematic Paleontology

TEMNOSPONDYLI Zittel, 1887–1890	
STEREOSPONDYLI Zittel, 1887–1890	
CAPITOSAURIA Yates & Warren, 2000, sensu Schoch, 2008	
CAPITOSAUROIDEA Säve-Söderbergh, 1935 sensu Schoch, 2013	

Yuanansuchus Liu & Wang, 2005

Revised diagnosis (modified from Liu & Wang, 2005)—Postorbital portion of skull occupied more than 1/3 of skull length; tabular horn directed laterally; lateral line sulci continuous, well impressed; supraorbital sensory canal enters lacrimal; preorbital projection of jugal shorter than half length of snout; vomerine plate short; interclavicle wider than long; ventral part of blade of clavicle with convex medial margin.

Yuanansuchus maopingchangensis sp. nov.

Etymology—From “Maopingchang”, the name of a village near the quarry.

Holotype—IVPP V 22628, a nearly complete skull (Fig. 3).

Type Locality and Horizon—Maopingchang, Yuanan County, Hubei Province, China; Member II of the Badong Formation, Anisian, Middle Triassic (Figs. 1 and 2).

Diagnosis—Skull elongated, orbit large; otic notch closed; frontal extends to level of posterior orbital margin; fodia vomeris absent; cultriform process of parasphenoid extends to level of anterior margin of interpterygoid vacuities.

Referred specimens: IVPP V 22629, a nearly complete skull (Fig. 4); IVPP V 22630, an incomplete skull (Fig. 5); IVPP V 22631, an incomplete left lower jaw (Fig. 5); IVPP V 22632, a right clavicle articulating with the interclavicle (Figs. 6A and 6B); IVPP V 22633.1-2, two interclavicles (Fig. 6C); IVPP V 22634.1-8, eight clavicles including at least five right ones (Fig. 6D).

Description

The following description, except where specified otherwise, is based on the holotype; the other two known skulls are used as a supplementary source of information.

The holotype is a slightly distorted skull missing the posterolateral corner of the right cheek (Fig. 3). As preserved, the skull has an anteroposterior midline length of 275 mm and a width of 215 mm, but the complete skull was probably about 23 cm wide. This skull is slightly longer than IVPP V 13463, the holotype of Yuanansuchus laticeps, but is also much narrower. The posterior part of the skull was exposed and damaged by weathering prior to excavation of the rest of the specimen, so that the occipital and otic regions are poorly preserved. The skull roof is generally well preserved, but white plaster has been inserted between the skull roof and palate to reinforce the skull. None of the marginal or palatal teeth are preserved.

Figure 3 Holotype.

Photos and drawings of holotype of Yuanansuchus maopingchangensis sp. nov. (IVPP V 22628) in (A) dorsal, (B) ventral and (C) occipital views. Abbreviations: apv, anterior palatal vacuity; ch, choana; cmp, crista muscularis of parasphenoid; co, crista obliqua of pterygoid; cp, cultriform process of the parasphenoid; Ec, ectopterygoid; Eo, exoccipital; F, frontal; fsm, fossa subrostralis media; ipf, interpremaxillary foramen; is, infraorbital sulcus; J, jugal; js, jugal sensory canal; L, lacrimal; lf, lacrimal flexure; M, maxilla; N, nasal; P, parietal; Pl, palatine; Po, postorbital; Pof, postfrontal; Pp, postparietal; Prf, prefrontal; Ps, parasphenoid; Pt, pterygoid; Q, quadrate; Qj, quadratojugal; spt, socket of palatine teeth; Sq, squamosal; ss, supraorbital sulcus; St, supratemporal; svt, socket of vomerine teeth; T, tabular; ts, temporal sulcus; V, vomer.

Figure 4 IVPP V 22629.

Skull of Yuanansuchus maopingchangensis sp. nov. (IVPP V 22629) in (A) dorsal, (B) ventral and (C) occipital views.

Figure 5 IVPP V 22630 and IVPP V 22631.

Skull of Yuanansuchus maopingchangensis sp. nov. (IVPP V 22630) in (A) dorsal and (B) occipital and (C) posterolateral views; lower jaw (IVPP V 22631) in (D) lateral, (E) medial and (F) dorsal views. A, angular; af, adductor fossa; Art, articular; amf, anterior meckelian foramen; D, dentary; Eo, exoccipital; f.ix-x, foramen for glossopharyngeal and vagal nerves; gf, glenoid fossa; hp, hamate process of the prearticular; Mc, middle coronoid; ms, mandibular sulcus; Par, preaticular; Pc, posterior coronoid; pmf, posterior meckelian foramen; Psp, postplenial; ptf, posttemporal fenestra; S, stapes; Sa, surangular; Sp, splenial; T, tabular; t, tooth.

Figure 6 Interclavicle and clavicle.

Interclavicle and right clavicle of Yuanansuchus maopingchangensis sp. nov. (IVPP V 22632) in (A) ventral and (B) dorsal views; (C) interclavicle (IVPP V 22633.1) in ventral view; right clavicle (IVPP V 22634.1) in (D) ventral and (E) dorsal views. Photo A was shot in the field.

In dorsal and ventral views, the outline of the skull is parabolic and elongate, with straight lateral margins. The outline is very similar to that of the skull of Quasicyclotosaurus campi (Schoch, 2000). The preorbital region is low in profile and relatively broad. The snout is long and slender compared with that of Y. laticeps. The two sides of the skull are asymmetrical. This anomaly can only partly be explained by distortion, and it seems likely that the preserved shape is similar to the original one. The midline suture between the frontals lies considerably further to the left than that between the nasals, so that the left nasal has a substantial contact with the right frontal. In most capitosaurs, the midline is nearly straight. Other differences in suture pattern exist between the left and right sides of the skull; for example, the right frontal is distinctly longer than the left one.

The dermal sculpturing is well preserved on the skull roof (Fig. 3) and largely consists of a pattern of deep pits, which grade into alternating ridges and grooves near the lateral margins of the skull and the nasal surface. The lateral line sensory sulci are continuous and obvious on the skull roof. The infraorbital sulcus (IS) has a Z-shaped lacrimal flexure, extends backwards along the lateral skull margin on the maxilla, and seems continuous with the jugal sulcus (JS) to the squamosal. The supraorbital sulcus (SS) is better-defined than the IS, runs across the prefrontal and the lacrimal, nearly contacts the IS, and extends onto the nasal. The SS seems to not meet its counterpart across the midline within the premaxilla. Posteriorly, the SS extends across the frontal and postfrontal, and intersects with the temporal sulcus (TS) on the postorbital. The TS extends downwards on the jugal to meet the IS, and extends posteriorly onto the supratemporal.

Skull roof

A small interpremaxillary foramen lies on the midline, close to the anterior border of the premaxillae. The nares are elongated and large (the left measuring 314 mm in length, and the right 275 mm), and are situated near the anterior margin of the skull. The external naris is bordered anteriorly and laterally by the premaxilla, dorsally by the nasal, and posteriorly by the maxilla. The posterolaterally directed suture between the premaxilla and the nasal touches the medial margin of the external naris, and the right suture is displaced anteriorly relative to the left one. In the dorsal view, the maxilla is restricted to the skull margin and does not extend posteriorly to the anterior margin of the orbit. No septomaxilla is evident. The nasal contacts the maxilla posterior to the external naris. The lacrimal contacts neither the orbit nor the naris.

The two large orbits are both ovoid, but differ in size, the right orbit being longer (43 mm in length) and more anteriorly positioned than the left (38 mm in length). Their sizes are slightly larger than those of Y. laticeps, which are 35 mm in length. However, both orbits are situated relatively far anteriorly, lying close to the mid-length of the skull roof. In this characteristic Y. maopingchangensis resembles Y. laticeps, Mastodosaurus giganteus and some juvenile capitisauroids (such as Watsonisuchus aliciae) (Liu & Wang, 2005; Schoch, 1999; Warren & Hutchinson, 1988). The orbits are almost dorsally directed and are raised above the rest of the skull roof, indicating an aquatic or semiaquatic lifestyle in which the animal habitually positioned itself near the water surface. The highest point on the skull roof is located on the postfrontal, just posterior to the orbit. The prefrontal is relatively long (measuring about one quarter of the midline length of the skull), tapers to an anterior point, and is elevated just anterior to the orbit. The frontal participates in the orbital margin, as is the case in many other capitosaurs (Damiani, 2001; Schoch, 2008) but not in Y. laticeps (Liu & Wang, 2005). The frontal is subequal in length to the nasal, although the left frontal is shorter than the right. However, in contrast to the condition in all other capitosaurs, the frontal extends posteriorly beyond the posterior margin of the orbit (Schoch, 2008; Schoch & Milner, 2000). The postfrontal is a nearly quadrangular in shape, and forms the posteromedial margin of the orbit. The postorbital is wide, and its lateral margin extends beyond that of the orbit. However, the postorbital does not extend as far anterolaterally as in many other capitosaurs, but narrows anteriorly and makes only a small posterior contribution to the orbital margin. The jugal forms most of the lateral border of the orbit, and has a long anterior process that extends far anteriorly to the orbit and reaches nearly the same level as the tip of the prefrontal. However, the length of the preorbital part of the jugal is nevertheless less than half that of the snout. Although the suture between the jugal and quadratojugal is unclear, the former is definitely one of the longest bone in the skull roof.

The skull table is very long for the anteriorly positioned orbit. The ratio of postorbital portion to the whole skull is 0.38. This ratio is similar to the ratio in Y. laticeps, bigger than in Mastodonsaurids (∼0.3), much bigger than in other capitosaurids (<1/4) (Liu & Wang, 2005; Fortuny, Galobart & De Santisteban, 2011; Schoch & Milner, 2000; Sidor, Steyer & Hammer, 2014). The parietal foramen is round and small, with a diameter of 6.5 mm, and lies on the midline near the anteroposterior midpoint of the parietals. The parietal is short, and is similar in length to the postparietal, the supratemporal, and the tabular. The postparietal is relatively long, its length exceeding its width. The posterior margin of two postparietals is incompletely preserved but looks transversely aligned as in Y. laticeps and Mastodonsaurus giganteus (Liu & Wang, 2005; Schoch, 1999). The tabular is short and wide, and its lateral process is only slightly posteriorly directed. As a result, the posterior margin of the skull roof is nearly straight in dorsal view as in Y. laticeps, rather than distinctly concave as in most capitosaurs (Schoch, 2008). The tabular contacts the squamosal, enclosing an otic fenestra and preventing the supratemporal from entering the dorsal margin of this opening. The otic fenestra is more or less rounded as in Eocyclotosaurus, not ovoid as in Quasicyclotosaurus and Cyclotosaurus; it embays the squamosal more deeply than the tabular as in Quasicyclotosaurus, Eocyclotosaurus, and Cyclotosaurus mordax (Schoch, 2000; Schoch & Milner, 2000). The squamosal is a large bone in the posterolateral part of the skull roof. The posterior margin of the squamosal is poorly preserved, but this bone does extend far enough posteriorly to overhang the occiput. The posterior margin of the postparietal, tabular and squamosal appears thick (∼7 mm) relative to the total height of the occiput (47 mm).

Palate

The palate is moderately vaulted, with the lateral edges slightly below the level of the middle portion. The maxilla bears more than 30 marginal tooth sockets, but no teeth are preserved. The anterior palatal vacuities are paired, oval in outline and completely separated by the premaxilla and the vomer, as in Eocyclotosaurus wellesi, Mastodonsaurus giganteus, and Paracyclotosaurus crookshanki (Damiani, 2001; Schoch, 1999; Schoch, 2000). A small fossa lies on the midline, crossing the suture between the premaxilla and the vomer. This structure may be a fossa subrostralis media, a feature present in Eocyclotosaurus woschmidti (Kamphausen, 1989).

The vomerine plate is longer than wide. The plate is in contact with the premaxilla anteriorly, the maxilla laterally, and the palatine posterolaterally, and it sends out a long posteromedial process that runs lateral to the cultriform process. The vomerine tusk pair is large, and situated at the anterolateral corner of the vomer. A triangular depression lies on the midline, behind the fossa subrostralis media. The transvomerine tooth row is not preserved, but the protruding base shows it is slightly convex posteriorly for the presence of the depression as in Edingerella madagascariensis (Maganuco et al., 2009). However, this depression is not observed in smaller specimen IVPP V 22629. The left side of the plate is poorly preserved, but on the right side a depression is situated posterior to the transvomerine tooth row.

The choana is elliptical in outline, and relatively long, as in most capitosauroids. It is bordered by the maxilla laterally, the vomer medially, and the palatine posteriorly. The interpterygoid vacuities are well-developed, measuring nearly 50% of the midline length of the skull, and are approximately as wide posteriorly as anteriorly. Each vacuity is bordered laterally by the palatine and pterygoid, and medially by the vomer and parasphenoid. The orbits are located at the level of the posterior half of the vacuities, as in Yuanansuchus laticeps. Apart from a small portion of the right orbit, they can be seen through the vacuities in their entirety.

The palatine forms a slender posterior process that meets the ectopterygoid laterally and the pterygoid posteriorly. The palatine tusk pair is not preserved, and even the shapes of their alveoli are unsure. The right ectopterygoid is a slender elongate bone situated medially to the maxilla. Shallow tooth sockets lie on the lateral margin of the bone, medial to the maxillary teeth. However, the part of the palate forming the lateral margin of the left interpterygoid vacuity is so mediolaterally compressed that even the suture between the maxilla and the ectopterygoid is unclear.

Although the middle portion of the vomerine plate is not well-preserved, the length of the midline suture between the two vomers shows that the parasphenoid extends anteriorly to around the level of the anterior margin of the interpterygoid vacuities. The anterior tip of the parasphenoid is far more posteriorly positioned than in most Capitosaurians, including Y. laticeps. The parasphenoid seems smoothly sutured with the vomer, and the fodina vomeralis should be absent here, in contrast to Yuanansuchus laticeps, The cultriform process is a flat bar which narrows laterally in the vicinity of its midpoint, and whose triangular base is smoothly continuous with the vaulted basal plate of the parasphenoid. The width of the basal plate is subequal to its length. The crista muscularis is a transversely oriented ridge located on the posterior rim of the basal plate. The ventrally positioned ‘pockets’ mentioned by Watson (1962) are indistinct and widely separated. The parasphenoid contacts the pterygoid laterally at an anteroposteriorly long suture, the length of the suture is slightly greater than the width of the corpus of the parasphenoid.

The palatine ramus of the pterygoid tapers anteriorly, and contacts the posterior margin of the palatine along a short suture located roughly at the level of the anterior margin of the orbit. The corpus of the pterygoid curves ventrally towards its lateral side; its lateral margin is granular and forms a roughened area. The quadrate ramus has a ventral oblique surface and an tall dorsal oblique ridge (crista obliqua), and extends posterolaterally to form a sutural contact with the main body of the quadrate. The palatine and quadrate rami of the pterygoid participate in medially delimiting the subtemporal fenestra, which is bordered posteriorly by the quadrate and laterally by the quadratojugal and maxilla. Both quadrates are preserved. This bone is nearly triangular in palatal view, and the medial side which contacts the pterygoid is much wider than the lateral side which contacts the quadratojugal. The ventral condyle for the articular is saddle-like. The quadrate bears a boss (hyoid tubercle) above the medial part of the condyle. The occipital condyles are separated from one another and are formed by the exoccipitals, which are in sutural contact with the parasphenoid anteriorly. The occipital condyles are positioned posteriorly to the level of the quadrate condyles. A long, slender, unidentified piece of bone is preserved on the left ventral margin of the skull. It could be part of a rib. The anterior end of this element covers the posterior portion of the upper alveolar row.

The skull of IVPP V 22629 is nearly complete, with a midline length of 21 cm and a width of approximately 15 cm (Fig. 4). The long axis of the orbit measures 27 mm, and the short axis measures 20 mm. Another skull (IVPP V 22630) is slightly smaller (Fig. 5). Both specimens are similar to the holotype in their general features, but differ from the holotype in that the frontal merely comes close to the orbit rather than entering the orbital margin. The frontal extends only to the level of the posterior margin of the orbits, rather than continuing posteriorly beyond this level as in the holotype. In IVPP V 22629, only one vomerine tusk socket lies anterior to the choana, and there is no socket lying posterior to the choana (Fig. 4B). In IVPP V 22629, the area of the contact between the vomer and the cultriform process is poorly preserved, and the anterior extent of the cultriform process is uncertain.

The occipital region is better preserved in two specimens. The occiput is relatively low because the occipital condyle is nearly situated at the same dorsoventral level as the quadrate condyle, as in Yuanansuchus laticeps (Liu & Wang, 2005). The posttemporal fenestra is triangular. The foramen magnum is wide. The right stapes is preserved in situ in both specimens, and the one in IVPP V 22630 is complete. The stapedial footplate is large, and the stapedial shaft is slab-like as in Tatrasuchus wildi (Schoch, 1997). The foramen for nerves IX–X (the glossopharyngeal and the vagal nerves) is visible on the lateral side of the exoccipital.

Lower jaw

IVPP V 22631 is a robust and long left lower jaw, missing the anterior tip and the postglenoid area (PGA) (Fig. 5). The intact specimen is estimated to have measured 27 cm in length and 65 mm in height. The preserved anterior tip of the ramus (approximately the midway between the mandibular symphysis and the forward edge of the adductor fossa) has a height to width ratio of approximate 1.5. The height to width ratio immediately behind the postglenoid ridge approximately is 1.7. These ratios are similar to those of Stanocephalosaurus birdi (Jupp & Warren, 1986). The lower jaw increases in height posteriorly until the posterior end of the coronoid, then begins to decrease in height on the lateral side of the adductor fossa. As a result, the dorsal margin of the lateral wall of the adductor fossa is strongly convex. The lateral surface of the angular shows well-developed radial ornamentations, which become less evident on the postsplenial. A distinct mandibular sulcus extends posterodorsally across the lateral surface of the angular, beginning at the ventral margin.

The dentary bears more than 20 tooth sockets, which are anteroposteriorly compressed. Only two broken teeth are preserved on the dentary. The middle and posterior coronoids both bear a single row of teeth, but only a few conical teeth are actually preserved. The medial side of the postsplenial bears a small anterior Meckelian foramen, posterior to which is an additional tiny foramen. The ratio of the length of anterior Meckelian foramen to the adductor fossa length is less than 0.1. The posterior Meckelian foramen is bordered by the postsplenial anteriorly and ventrally, the prearticular dorsally, and the angular posteroventrally. It is elongated and much larger than the anterior Meckelian foramen, but still far less than half as long as the adductor fossa. The center of posterior Meckelian foramen lies anterior to the vertical line passing through the anterior-edge of the adductor fossa. The medial wall of the adductor fossa is incomplete, but the original height of medial wall could be deduced from preserved portion. The height of the medial wall should be slightly greater than 2/3 height of the lateral wall at the middle of the adductor fossa length. The prearticular extends anteriorly beyond the anterior margin of the posterior Meckelian foramen, and contacts the postsplenial below the posterior coracoid. The hamate process of the prearticular is not complete, but still rises above the level of the glenoid fossa. Most of the glenoid facet lies above the level of the dorsal surface of dentary.

Although the postglenoid area is nearly completely missing, the following features on this area still can be deduced from the preserved specimen: the prearticular does not extend into the PGA, and the angular does not participate in the PGA. So the PGA should be type I of Jupp & Warren (1986) for this specimen.

Postcranial skeleton

Many isolated bones have been discovered at this locality, but only a limited number have been successfully excavated because they are quite fragile and the matrix is very hard. Among the elements that have been recovered, three interclavicles and nine clavicles have been identified as temnospondyl in origin and tentatively referred to Y. maopingchangensis (see ‘Discussion’).

The interclavicle is well preserved in IVPP V22632 (Fig. 6). It is rhomboidal in shape, lacking a distinct anterior stylus, and is much wider than long. In contrast, the interclavicle is longer than wide in Mastodonsaurus giganteus (Schoch, 1999). On the dorsal side, this bone bears three converging low ridges (Figs. 6B and 6C). The anterior two ridges lie in the anteroposteriorly middle portion of the bone, and the posterior one is longitudinally aligned. The anterior and posterior margins of the interclavicle are broad and nearly straight. The ventral surface displays the characteristic temnospondyl dermal sculpture pattern, with furrows and ridges radiating outwards from the ossification center.

Among the nine clavicles, only two right clavicles are nearly complete (Fig. 6). The ventral blade of the clavicle is slightly longer than wide, and shows a radial ornamentation on the ventral surface. The medial margin is convex, making the blade fanlike in shape rather than triangular. The dorsal process is posterodorsally directed and similar to that of Stanocephalosaurus pronus (Schoch & Milner, 2000; Howie, 1970). The lateral surface of the dorsal process combines with the ventral blade to form a prescapular cavity, and the dorsal process tapers to form a sharp dorsal tip.

Discussion

The specimen IVPP V 22628 should represent an adult stage of the animal for the large size of the skull, the oval cross-section of the marginal teeth, ornament of coarse pits extending to sutural boundaries, well-ossified exoccipitals and quadrates (Steyer, 2003; Warren & Hutchinson, 1988). Specimens IVPP V 22632, 22633.1, and 22634.1 also should represent an adult stage for central reticulated ornamentation on their ventral surfaces, turning into radial ornamentation at their peripheries (Steyer, 2003).

The only previously reported tetrapod species from the Maopingchang locality is Yuanansuchus laticeps. The specimen IVPP V 22628 is different from the only known specimen of Yuanansuchus laticeps, and cannot be referred to this species. IVPP V 22628 differs from Y. laticeps in possessing the following features: elongated skull, large orbit, frontal enters the medial border of the orbit, closed otic fenestra, absence of fodina vomeralis and cultriform process of parasphenoid extends to the level of the anterior margin of the interpterygoid vacuities. Particularly considering that IVPP V 22628 is similar in skull length to the holotype of Y. laticeps, the differences between the two specimens are too great in magnitude to represent intraspecific variation, implying that IVPP V 22628 represents a new taxon. This specimen has a laterally directed tabular horn, as in the holotype of Y. laticeps, and the lateral orientation of the horn is a unique feature not known in other capitosaurs. Although such capitosaurs as Eryosuchus and Mastodonsaurus have been described as having laterally directed tabular horns (Damiani, 2001; Schoch, 2008), their tabular horns in fact differ from those of the two Chinese taxa in being directed posterolaterally.

The two small skulls collected from the same small hill (IVPP V 22629, 22630) are very similar to IVPP V 22628 in most features but differ in that the frontal is excluded from the orbital margin in both specimens. As shown by Kamphausen (1994), however, frontal asymmetry can be present even in a single skull, so these two specimens are referred to Y. maopingchangensis here. The lower jaw and postcranial bones are tentatively also referred to Y. maopingchangensis because they came from the same small spot and same horizon where the skulls described in this paper were found. All of the cranial material from this spot is referable to Y. maopingchangensis. Because postcranial morphology is relatively stable within most small temnospondyl clades although the ossification varies during ontogeny (Pawley, 2007; Pawley & Warren, 2005), the postcranial features observed in the new Maopingchang specimens probably characterize the entire genus Yuanansuchus rather than only Y. maopingchangensis.

Phylogenetic analysis

To assess the phylogenetic position of Yuanansuchus maopingchangensis, I incorporated this taxon into the data matrices of Schoch (2008) and Sidor, Steyer & Hammer (2014), both of which represent modifications of that of Damiani (2001). The original dataset of Damiani (2001) included 47 characters. Fortuny, Galobart & De Santisteban (2011) analyzed a modified version of this matrix, which excluded Damiani’s (2001) characters 22 and 34 but added eight new characters. Subsequently, Sidor, Steyer & Hammer (2014) added Antarctosuchus to the matrix of Fortuny, Galobart & De Santisteban (2011). When I ran the matrix of Sidor, Steyer & Hammer (2014) using the same software (PAUP 4.0 beta 1.0 for PC) (Swofford, 2001) and settings specified in the paper, the results differed from those reported by Sidor, Steyer & Hammer (2014) in that A. polyodon was found to be the sister taxon of Mastodonsaurus giganteus rather than that of Paracyclotosaurus crookshanki, although the parameters two equally most parsimonious trees were the same (145 steps, a consistency index of 0.40, and a retention index of 0.70). Except with respect to the position of Antarctosuchus, however, the two trees are topologically identical with each other and with the cladogram published by Fortuny, Galobart & De Santisteban (2011). However, this could be due to the type error for the coding of character 50. If the coding is changed to 1, the result is same as the original paper. The two species of Yuanansuchus were added to this dataset (see Appendix S1), which was analyzed in PAUP 4.0 beta 1.0 for PC using the same settings as previous analyses. This analysis recovered four most parsimonious trees, the strict consensus of which is almost identical to Fig. 7 of Sidor, Steyer & Hammer (2014) except the position of Chinese clade comprising Y. laticeps and Y. maopingchangensis (Fig. 7).

Figure 7 Cladogram.

Cladistic relationships of Yuanansuchus within capitosaurian temnospondyls recovered from a cladistic analysis of 29 taxa and 53 characters, strict consensus of four equally most parsimonious trees of 154 steps. See Appendix S1 for the data.

Schoch (2008) revised the characters of Damiani (2001) and added 19 further characters and used a slightly different list of taxa. He obtained results quite different from those of Fortuny, Galobart & De Santisteban (2011) and Sidor, Steyer & Hammer (2014), one key difference being that Edingerella was recovered as a trematosaur rather than a capitosaur by Schoch (2008). However, when Schoch’s (2008) small matrix of 25 taxa was analyzed in PAUP 4.0 beta 1.0 for PC, the shortest trees were found to be only 152 steps in length, whereas the single most parsimonious tree reported by Schoch (2008) was 162 steps. The strict consensus of the trees obtained is similar to the topology depicted in Schoch’s (2008) Fig. 8, but includes the monophyletic clade (((Cyclotosaurus (Eocyclotosaurus, Quasicyclotosaurus)), Mastodonsaurus), Paracyclotosaurus). The matrix of Schoch (2008) including Yuanansuchus laticeps resulted in an unresolved strict consensus tree.

Figure 8 Cladogram.

Cladistic relationships of the capitosaurian temnospondyls based on 29 taxa and 56 characters. Three most parsimonious trees (A, B, C) were recovered from the analysis when Yuanansuchus maopingchangensis is coded for all known specimens, while two of them (A, B) were obtained when Y. maopingchangensis is coded for only holotype.

To resolve the interrelationships of Capitosauria, a new data matrix was constructed using a revised list of 56 characters (Appendix S1). The codings of the characters were mostly adopted from previous analysis, though the matrix incorporates a few novel codings based on examination of specimens, photos and the literature (Table 1). Y. maopingchangensis was coded primarily based on the holotype, and secondarily with reference to the other specimens described in this study. The matrix was analyzed with TNT 1.1 (Goloboff, Farris & Nixon, 2008), and all characters were equally weighted and left unordered. The holotype only dataset recovered two most parsimonious trees with 185 steps in length; while the composite dataset recovered three most parsimonious trees with 186 steps in length, two have same topology as the previous (Fig. 8). One tree (Fig. 8A) is similar in some degree to its topology to that depicted in Fig. 7A of Fortuny, Galobart & De Santisteban (2011). Both topologies posit the two Chinese taxa as a monophyletic clade, supporting referral of the new species to Yuanansuchus. This clade is supported by the following synapomorphies in all shortest trees: (character 4) laterally directed tabular horn; (character 7) lateral line sulci continuous, well impressed; (character 10) supraorbital sensory canal entering larimal; (character 13) preorbital projection of jugal shorter than half the length of snout. Yuanansuchus does not form a clade with Eocyclotosaurus plus Quasicyclotosaurus as suggested by Liu & Wang (2005). The positioned recovered for this clade is more basal than that recovered for Y. laticeps by Liu & Wang (2005) and Schoch (2008), but still falls on the branch of capitosauroid phylogeny that includes Cyclotosaurus. Schoch (2013) defined Capitosauroidea as the least inclusive clade containing Parotosuchus nasutus and Cyclotosaurus robustus. Here, Capitosauroidea can be divided into two clades: a relatively small one containing Parotosuchus, and a much larger one containing Cyclotosaurus. Capitosauroidea includes Edingerella, Watsonisuchus, and Xenotosuchus in two most parsimonious trees (Figs. 8A and 8C) but not in the other (Fig. 8B). In the phylogenies of Fortuny, Galobart & De Santisteban (2011) and Sidor, Steyer & Hammer (2014), Parotosuchus and Cyclotosaurus are positioned close to one another, resulting in a restricted Capitosauroidea that excludes Paracyclotosaurus and many other taxa recovered as capitosauroids in the present analysis. The present analysis did not recover a close relationship between Eryosuchus and Mastodonsaurus, and Antarctosuchus polyodon is the sister taxon of Eocyclotosaurus plus Quasicyclotosaurus rather than Paracyclotosaurus, supported by (character 31) quadratojugal excluded from quadrate trochlea.

Table 1 Data matrix.

Data matrix used in the phylogenetic analysis.

Angusaurus spp.	1110011111	0100111011	1121100010	0010011010	1111100000	100100	
Benthosuchus sushkini	1100101101	0101110011	0110100011	0011011100	0011000000	100100	
Calmasuchus acri	0101000?00	11?1?1?102	00001?1011	111????112	0011000001	011??1	
Cherninis denwai	0101100200	1112010000	0000111011	11111111??	?????0000?	100010	
Cyclotosaurus robustus	0101000200	1111001200	1010111111	1111111122	10???10110	002111	
Edingerella madagascariensis	010010[01]200	1111100001	00001[01]1010	10010?1101	001?011000	000100	
Eocyclotosaurus spp.	1101101201	0112111002	1110111111	10111111?2	?????10010	122111	
Eryosuchus garjainovi	010110020[01]	111101110[01]	0000111011	1111111122	1011000101	000000	
Lydekkerina huxleyi	0110000100	0101000002	0000100001	0010001010	0011000000	000000	
Mastodonsaurus giganteus	0101001201	1101011102	1010111111	1111111122	1111000110	100011	
Odenwaldia heidelbergensis	110010020?	?1110?0??2	????????1?	??111?11??	?????00???	100?1?	
Paracyclotosaurus crooshkanki	0101100200	1112010102	100?111111	11111111??	?????1111?	11101?	
Parotosuchus orenburgensis	0100100200	1112010100	0010111011	1111111111	0011000001	100100	
Procyclotosaurus stantonensis	1101100200	111201100?	0???1??111	?1111111??	?????0111?	1[12]1?11	
Quasicyclotosaurus campi	0101101201	0102101002	001011?111	10??111111	10???10110	122111	
Rhineceps nyasaensis	0000000000	0001000000	0000000000	0000000100	0000000000	00010?	
Tatrasuchus wildi	0101?00200	1111000200	0000111011	1111111111	1111010111	001111	
Thoosuchus yakovlevi	1100011111	0100110012	0121100000	0010011000	1111100000	100100	
Trematosaurus brauni	1110011111	01001110?2	1121???010	01100?1020	1111100000	100100	
Stanocephalosaurus birdi	0101000200	1112011 001	0001111111	1111111111	0011011011	111001	
Stanocephalosaurus pronus	0101000200	1112011000	0001111111	1111?11111	00?1011010	111001	
Uranocentrodon senekalensis	0000000000	0000000000	0000000001	0?00000100	0000000000	000000	
Vladlenosaurus alexeyevi	1100100200	0110010010	01001?1011	1???????11	?11?000000	100111	
Watsonisuchus spp.	0100100200	1112010010	0000111011	1011111111	0011011000	100100	
Wetlugasaurus angustifrons	010010020[01]	0111010010	0100111011	0101111111	1011000000	100100	
Xenotosuchus africanus	0101100200	1111011000	0000111011	11?1111111	0011011100	10010?	
Antarctosuchus polyodon	0?0??012?0	111201110?	10?0110111	10??1111??	???????11?	11?101	
Yuanansuchus laticeps	0102101211	010100100?	0000111011	1?111111??	?????1100?	2010?1	
Yuanansuchus maopingchangensis	0102101201	1101101002	0000111011	11?11111??	?????110??	102100	
Yuanansuchus maopingchangensis	0102101201	[01]101101002	0000111011	11?11111?1	01110110??	102100	
Notes.

The taxa in the matrix include all 26 species analyzed by Fortuny, Galobart & De Santisteban (2011), plus Antarctosuchus, Yuanansuchus laticeps and Y. maopingchangensis. Codings for most species were adapted from Schoch (2008) and Fortuny, Galobart & De Santisteban (2011), but codings for Antarctosuchus were adapted from Sidor, Steyer & Hammer (2014) and the two species of Yuanansuchus were coded based on personal observation of specimens. Y. maopingchangensis was coded primarily based on the holotype, and secondarily with reference to other known specimens. A few previous codings are revised here based on photos and the literature. Character 4 is redefined and following the coding of Schoch (2008). Character 49 here was coded as (0) in Antarctosuchus by Sidor, Steyer & Hammer (2014) but changed to (1) here. The coding of character 35 is changed from 0 to 1 for Quasicyclotosaurus following Schoch (2008) and for Y. laticeps.

Conclusion

The new temnospondyl specimens collected from the Maopingchang site (Yuanan, Hubei, China) are assigned here to a new capitosaur species, Yuanansuchus maopingchangensis. This species is the second to be named from the site, and shares the following features with Y. laticeps: tabular horn directed laterally; lateral line sulci continuous, well impressed; supraorbital sensory canal enters lacrimal; preorbital projection of jugal less than half length of snout; and vomerine plate short. Y. maopingchangensis differs from Y. laticeps in having the following characteristics: elongated skull; large orbit; frontal that may enter medial border of orbit; closed otic fenestra; and short cultriform process. A phylogenetic analysis confirms that the two species form a monophyletic clade, and supports the existence of a major dichotomy between separate Parotosuchus and Cyclotosaurus lineages within Capitosauroidea.

Supplemental Information

Appendix S1 Character list

Click here for additional data file.

I thank my field team (Li Lu, Jia Zhenyan, Xu Xu) for collecting the specimens, Wu Yong and Fu Hualin for preparing the specimens, Xu Yong for helping to illustrate the paper. Corwin Sullivan kindly read the draft of this paper and significantly improved the language. Boris Morkovin (Borissiak Paleontological Institute, Moscow, Russia) kindly sent some photos and references. I thank Jean-Sébastien Steyer, Josep Fortuny, and Tomasz Sulej for their comments and suggestions.

Abbreviations

IVPP Institute of Vertebrate Paleontology and Paleoanthropology, Chinese Academy of Sciences, Beijing, China.

Additional Information and Declarations

Competing Interests

Author Contributions

Data Availability

New Species Registration

The author declares there are no competing interests.

Jun Liu conceived and designed the experiments, performed the experiments, analyzed the data, contributed reagents/materials/analysis tools, wrote the paper, prepared figures and/or tables, reviewed drafts of the paper.

The following information was supplied regarding data availability:

The raw data is supplied in Appendix S1 and the manuscript.

The following information was supplied regarding the registration of a newly described species:

Publication LSID: urn:lsid:zoobank.org:pub: D68F2D09-0AC0-4AE1-ACE1-EFD0C5680916

Y. maopingchangensis LSID: urn:lsid:zoobank.org:act:52627651-9232-439B-8BB1-8312545450A9.

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
