# Peer review of "Yuanansuchus maopingchangensis sp. nov., the second capitosauroid temnospondyl from the Middle Triassic Badong Formation of Yuanan, Hubei, China"

_PeerJ, doi:10.7717/peerj.1903_

## Round 0.1 · original submission · Major Revisions

Based on the comments of two reviewers, the paper is of interest but requires major revisions. The author should carefully review and respond in detail to each point raised by the reviewers.

·

Basic reporting

The author describe new cranial and postcranial capitosauroid material from the Middle Triassic of Badong Formation, China.
The new material is well preserved and interesting and surely will help to elucidate the phylogenetic position of the genus Yuanansuchus.
The paper is well written but requires a major revision prior to its publication.
I made several annotations in the attached PDF documents (mainly in the main text, but also in the character list).
In my opinion the erection of the new species should be avoided as represents new specimens of the previously erected species Yuanansuchus laticeps from the same area (Liu and Wang, 2005). Anyway, the paper may be acceptable in this journal without the erection of a new species. The article is interesting because is probably revealing the intravariability of this taxon and probably morphological changes across the ontogeny.
On this aim, the author needs to discuss the ontogeny stage of the new material. The supposed holotype is a young adult or late individual? And the referred specimens? I’m wondering if the holotype of Yuanansuchus laticeps may represents a late young individual while the new material represents adult specimens from the same species. This could explain the unresolved phylogenetical position of Y. laticeps found in previous analysis (e.g. Schoch, 2008). May the situation is similar to the capitosauroid Edingerella madagascariensis: different taxa was erected and later joined in a unique taxon, because represent different ontogenetical stages (please, check: Steyer, 2003 JVP; Maganuco et al. 2009 Mem. His. Nat. Milano). Please discuss in deep this question in the discussion section.
Regarding the locality, the author claims that was found in different stratigraphical position (20 m below of Y. laticeps). However, no stratigraphic column or further geological information is included in this MS (neither presented in Liu and Wang, 2005). The author only includes a picture of the excavated locality without any other information. I think that it’s mandatory to include a stratigraphic column showing the different members of the Badong Formation and with special detail in Member II (where the capitosaur material was recovered), with precise information of the stratigraphic position of both localities. Any additional sedimentological information could be of interest.
About the material description, there are some morphological characters that need to be more confidently described and be more concise in some details. Please check my annotated document but just a couple of examples: A) Frontal contacting orbit: is this a polymorphic state or not? Intravariability? Saying that “may” enter is not scientifically consistent. B) Some valuable information should be added and/or discussed about the lower jaw, especially from the PostGlenoid Area (PGA).

Experimental design

The cladistics analysis needs further explanations and discussion.
I tested the cladistics matrix and I obtained the same results using Paup software.
However, there are some problems that needs to be solved:
I agree with the author that there is a problem in the matrix line published by Sidor et al. 2014 because it appears nested with Mastodonsaurus instead of Paracyclotosaurus. I checked the characters and I think that there is a typo mistake in the published matrix line in character 50: this character must be 1 instead of 0.
I repeated the analysis presented by Sidor et al. (2014) (changing ch. 50) and I obtained the same results published by Sidor et al. 2014: Antarctosuchus is nested with Paracyclotosaurus.
Thus, it could be interesting if in this new paper, the author solves this typo for ch.50 and explain it in this new publication.

The most important problem (of this section but probably of the whole paper) is about the codings of the characters and the interpretation of the results:
On one hand, the author claims to use the same character (and definition of the characters) than previously analysis (Damiani, 2001; Schoch, 2008; Fortuny et al. 2011) just mentioning in the table legend that some codings changed but not giving information about which codings changed and justification of the changes (specimen analyzed, literature checked…).
If the author made some changes in the codings, he must explain them (which ones, why, source data) to validate these results. If this information is not given the results could be not replicated for other workers.
I didn’t check exhaustively, but just an example:
Character 4: tabular horns: this is a key character used for several decades to elucidate the phylogenetical relations within the capitosaurs and is well known for most (if not all) taxa. In this paper the author claim to use the character defined by Damiani (2001) and followed by Schoch (2008) and Fortuny et al. (2011), but the codings for this character not fits with the codings of the original data matrix (e.g. character 4 is considered 0 for Angusaurus in Fortuny et al. 2011 and 1 in the present study. In similar way, Cyclotosaurus is considered 1 by the present study but coded as 2 for others.
Please justify each one of the modified codings, including literature source and first-hand analyzed specimens.
Just a suggestion for the author: I tested using the original data matrix of Fortuny et al. 2011 including the matrix lines of the author for the two supposed species of Yuanansuchus as well as Antarctosuchus (solving ch. 50) and I obtained exactly the same tree topology of Fortuny et al. 2011 with Antarctosuchus nested with Paracyclotosaurus and Yuanansuchus taxa being the sister taxon of Eocyclotosaurus+Quasicylotosaurus (forming part of the Heylerosaurids). This result is in agreement with Liu and Wang 2005, but different from the new results presented herein.
In this way: the author claims in line 371 that his results are similar in its basic pattern to Figure 7A of Fortuny et al. 2011. However, this is not true. There are several differences from the new results to the general pattern of Fortuny et al. 2011. Many taxa changed its position with important phylogenetical, chronographical and paleobiogeographical implications.
Please discuss if Yuanansuchus should be considered or not a member of the Heylerosaurids, closely related to Quasicyclotosaurus+Eocyclotosaurus as stated in Liu and Wang 2005. Also, it could be important also to discuss the paleobiogeographical implications of your results to evaluate the plausibility of the different scenarios.

Validity of the findings

No comments

·

Basic reporting

This is a very interesting discovery, as temnospondyls from China remain rare. The general structure of the article (=the plan) and the English are good, but the
systematics identification lacks anatomical comparisons, the diagnosis lacks autapomorphic characters, and the data matrix used for the phylogeny has been revised based on bibliography only. This very nice and new material deserves a more precise description (ex. the occipital view of the holotypic skull is not figured) and a specimen-based comparison (especially if the author wants to revise the data matrix for his phylogeny).

Experimental design

Arguments about the stratigraphic and somatic ages of the specimens are missing: please add a stratigraphical log as a figure, and add comments on the somatic/individual age based on comparisons, ornamentation pattern and ossification degree.

Validity of the findings

According to my knowledges on the capitosauroids, the specimens described here are too different than the holotype of Yuanansuchus laticeps to belong to the same genus. I invite the author to (re)consider the importance of the otic and postglenoid regions in the capitosauroid phylogeny, as demonstrated by the other previous analyses (cf. Damiani, Schoch's works).

Additional comments

I attach to this review my notes, corrections and suggestions directly written in the annoted PDF version of your manuscript. I remain at your disposal for any other review.

·

Basic reporting

No comments.

Experimental design

No comments.

Validity of the findings

No comments.

Additional comments

The paper is a description of the new species of capitosaurid amphibian with basic phylogenetic analysis. It is a pity that author did not present sens of evolutionary process of this interesting group and stopped on the simple cladistic analysis, but I understand that now it is standart way in "modern" paleontology.

---

## Round 0.2 · accepted · Accept

The referees' comments were adequately addressed in the revised version of the manuscript.